# Maternal Nutritional Status and the Microbiome across the Pregnancy and the Post-Partum Period

**DOI:** 10.3390/microorganisms11061569

**Published:** 2023-06-13

**Authors:** Katie M. Strobel, Sandra E. Juul, David Taylor Hendrixson

**Affiliations:** Department of Pediatrics, University of Washington School of Medicine, 1959 NE Pacific St., Seattle, WA 98195, USA

**Keywords:** maternal nutrition, obesity, pregnancy, microbiome

## Abstract

Appropriate nutrition during pregnancy and the post-partum period is vital for both the mothers and their offspring. Both under- and over-nourished status may have important microbial implications on the maternal and infant gut microbiomes. Alterations in the microbiome can have implications for a person’s risk of obesity and metabolic diseases. In this review, we examine alterations in the maternal gut, vaginal, placental, and milk microbiomes in the context of pre-pregnancy BMI, gestational weight gain, body composition, gestational diabetes, and maternal diet. We also investigate how the infant gut microbiome may be altered by these different parameters. Many of the microbial changes seen in under- and over-nourished states in birthing parents may result in long-term implications for the health of offspring. Differences in diet appear to be a major driver of the maternal and subsequently milk and offspring microbiomes. Further prospective longitudinal cohort studies are needed to examine nutrition and the microbiome to better understand its implications. Additionally, trials involving dietary interventions in child-bearing age adults should be explored to improve the mother and child’s risks for metabolic diseases.

## 1. Introduction

Pregnancy and post-partum periods are times of significant metabolic and microbial changes in the birthing parent and their offspring. There are increased energy requirements to support the growth of the uterus, breast, placenta, and fetal tissues [1]. The gut microbiome facilitates nutrient absorption, gut defense barrier, and immune development. There is an increase in gut *Akkermansia*, *Bifidobacterium*, and Firmicutes bacteria during pregnancy, which are likely to facilitate energy storage [2]. There is also an increased abundance of the gut bacteria Proteobacteria and Actinobacteria, which are believed to protect against inflammation [2]. The nutritional status of a pregnant person is important in regulating the microbial shifts necessary during pregnancy and the post-partum period.

Both excessive and inadequate nutrition can have remarkable consequences for mothers during pregnancy (Table 1). Obesity, excessive gestational weight gain, gestational diabetes, and hypertension have been associated with an increased risk of obesity, cardiovascular disease, and type 2 diabetes [1]. In contrast, inadequate nutrition during pregnancy has been associated with an increased risk of life-threatening hemorrhage, obstructed labor, sepsis, and all-cause mortality [3,4,5].

Excessive and inadequate nutrition can also have consequences for the offspring (Table 2). The effects of maternal nutritional status on infant outcomes may be mediated via several pathways, including dietary intake, milk composition, and maternal microbiome (Figure 1). Infants of mothers with gestational diabetes and/or pre-pregnancy obesity have an increased risk of increased fetal growth, large for gestational age status at birth, and later metabolic syndrome [20]. Inadequate nutrition during pregnancy alters placental histomorphology and function [21] and can lead to epigenetic changes in nutrient utilization, as well as a higher risk of fetal growth restriction, small for gestational age status, and later metabolic syndrome [21]. Additionally, human milk oligosaccharides (HMOs) can be altered by nutritional status [22,23] and are prebiotics for milk and gut bacteria. Thus, HMOs may alter the milk microbiome and shape the infant gut microbiome [24,25,26].

Alterations in the microbiome have been associated with inappropriate nutrition during pregnancy. Both obesity and undernutrition during pregnancy have been associated with decreased gut microbial diversity and shifts in microbial abundance [34]. However, changes in diet can modify the microbiomes of mothers and offspring. In this review, we will explore the impact of different markers of metabolic health and diet on the maternal and infant microbiome during pregnancy and the post-partum period.

## 2. Maternal Metabolic Health and Its Role in Microbial Changes

### 2.1. Overweight/Obesity

Increases in pre-pregnancy body mass index (BMI) have been associated with alterations in the maternal gut microbiome (Figure 2). In the first trimester, pregnant mothers who were obese prior to pregnancy were found to have a higher relative abundance of Firmicutes and a lower relative abundance of Proteobacteria compared to their normal body weight counterparts [6]. A high abundance of Firmicutes has been found in multiple studies of adults with obesity, with evidence suggesting that Firmicutes increase the efficiency of energy extraction and promote the absorption of calories [35]. In the third trimester, overweight pregnant mothers were found to have increased *Bacteroides* [7], *Clostridium* [7], *Biolphila* [8], *Roseburia* [8], *Dialster* [8], and *S. aureus* [7] and decreased *Phascolarctobacterium* compared to normal BMI mothers [8]. *S. aureus* has been found in the presence of intestinal inflammation secondary to adipocyte hyperplasia [35]. However, these changes are inconsistent across studies; one study found no changes in genus-level composition between those with an elevated BMI and those with a normal BMI [36]. Similarly, some studies have found decreased diversity with obesity [34], whereas others did not [8]. This could be due to differences between maternal cohorts regarding gestational weight gain, body composition, and maternal diet, which have all been found to be drivers of the maternal gut microbiome in the pregnancy and post-partum periods.

Obesity has also been associated with alterations in maternal vaginal and placental microbiomes. Alterations in the vaginal and placental microbiomes have been associated with preterm births [37]. Women with normal weight have increased diversity in the vaginal microbiome in the introitus and post-fornix compared to those with obesity [38]. Another study on Caucasian mothers found that pre-pregnancy BMI was associated with an elevated Nugent score, a score concerning vaginal dysbiosis [39]. The role of obesity in the placental microbiome has been controversial. Maternal obesity is believed to contribute to placental dysfunctions. In a pig model of obesity during pregnancy, the obese group showed increased oxidative damage with increased reactive oxygen species protein [40]. These reactive oxidative species and interleukins were correlated with a relative abundance of *Christensenellacaea_R-7* and decreased *norank_f_Bacteroidales_S24-7_group,* which are members of the Firmicutes phyla. These findings suggest that placental inflammation may be secondary to microbial changes and could contribute to the risk of perinatal morbidities, such as preterm birth. However, when examining the placentas of mothers who delivered preterm, there was no clustering of microbial communities by obesity, but there was clustering by gestational weight gain [41]. In contrast, a study in term mothers found that the placental microbiome of obese pre-pregnant mothers had less diversity, microbial richness, and abundance of taxa [42]. It is possible that changes in the placental microbiomes could be observed at an earlier gestation with excessive gestational weight gain rather than obesity, as obesity is a chronic disease process.

HMOs are considered a prebiotic for infant gut bacteria, and thus altered composition could impact the gut microbiome. The milk microbiome and HMOs may be altered in obesity (Figure 2). In general, mothers with an elevated BMI have been found to have a less diverse milk bacterial community [43]. An elevated BMI has been associated with a higher relative abundance of *Staphylococcus* [44,45,46], *Akkermansia* [43,44], *Corneybacterium* [46], and *Granulicatella* [46,47], and low abundance of *Lactobacillus* [44,45], *Bacteroidetes* [47], *Bifidobacterium* [43,44], and *Streptococcus* [45] in the milk microbiome. However, there are some studies that have found no relationship between BMI and milk microbial composition [48,49]. These differences in results could be due to differences in the maternal HMO concentrations between cohorts. One study found that obese mothers were more likely to be non-secretors (those who cannot produce fucosylated HMOs), and among non-secretors, there was increased sialyl-lacto-N-tetraose b (LSTb) and fucosyl-disialyl-lacto-N-hexaose (FDSLNH) in overweight mothers compared to normal-weight mothers [50]. Lacto-N-pentose II/III and lacto-N-fucopentose I are associated with maternal pre-pregnancy BMI [49]. Fucosylated HMOs are an important energy source for the infant gut *Bifidobacterium* and may affect the development of the infant gut [51].

Maternal overweight/obesity status has also been associated with alterations in the infant gut microbiome (Figure 2). One study found that meconium samples of infants born to mothers who were obese pre-pregnancy had less Firmicutes and increased Proteobacteria compared to those of infants born to mothers with normal pre-pregnancy BMI [6]. When examining infants of overweight mothers for up to 12 months, diversity indices of the gut microbiome were increased in the overweight group compared to that of the normal-weight group [27]. However, the evidence of changes in infant gut bacterial abundance is heterogenous with their findings, as they examine differing time points. In a study examining infant gut bacterial abundance at the genus level at 1 month and 6 months, infants from overweight mothers had a greater abundance of *Salmonella, Serratia,* and *Coprobacillus*; however, significance was only achieved with unadjusted *p*-values [27]. Another study found that a mode of delivery altered the findings. There were no associations between pre-pregnancy BMI and the infant gut microbiome at 6 weeks of age in the cesarean-delivered infants, but in the vaginal birth-delivered group, there was an increase in *Bacteroides fragilis, Escherichia coli*, *Veillonella dispar*, *Staphylococcus,* and *Enterococcus* [9]. Infants of overweight mothers had fewer butyric acid-producing bacteria compared to normal-weight mothers [27]. These findings suggest that infants of overweight/obese mothers begin with pathogenic bacteria with less butyrate production. Butyrate is considered the optimal energy source for colonocytes, and a limited source of this could impact intestinal health [52].

### 2.2. Underweight

The effects of maternal underweight habitus on maternal and infant microbiomes are less well-defined than those of maternal obesity and overweight habituses. There is animal data to suggest that maternal undernutrition alters the microbiome. A study of pregnant cattle demonstrated decreased placental microbial diversity in the setting of feed restriction during late gestation [53]. Furthermore, feed restriction in pregnant ewes results in an altered relative abundance of gut microbial communities [54]. Feed restriction in ewes additionally results in decreased colonic microbiome diversity, increased relative abundance of *Peptococcaceae*, and decreased relative abundance of *Ruminococcus* [55]. *Peptococcacea* is associated with higher high-density lipoprotein cholesterol [56], whereas *Ruminoccocus* plays a key role in complex carbohydrate degradation [57] and may be important in maintaining colonic health, as a decreased abundance of Ruminococcaceae has been implicated in inflammatory bowel diseases [10,58] and antibiotic-associated diarrhea [13].

Evidence from individuals with anorexia has demonstrated decreased alpha diversity [59] and lower amounts of total bacteria and obligate anaerobes in fecal samples when compared to well-nourished individuals [54]. A reduction in the energy balance and nutrient load in the diet results in an increase in Firmicutes and a decrease in Bacteroidetes, which is similar to that in obese adults [11]. This could be plausible as these bacteria facilitate energy usage, and in a starved state, this is critical.

Additionally, evidence from young children with severe undernutrition demonstrates less mature gut microbiomes that may promote excessive weight loss [12]. Children with malnutrition have lower richness and an increased abundance of Proteobacteria, including pathogenic *Klebsiella* and *Escherichia*, and a lower abundance of *Bacteroidetes* when compared to healthy children [60]. Reduced diversity and increased relative abundance of *Acidaminococcus* are also reported among children with undernutrition [14]. Taken together, it is plausible and likely that undernutrition during pregnancy results in distinct changes in the maternal microbiome, which will affect offspring, although additional studies focusing on the microbiome of undernourished individuals are needed.

### 2.3. Gestational Weight Gain

The role of gestational weight gain during pregnancy in the maternal gut microbiome is controversial. Gestational weight gain is often dependent on obstetrician recommendations due to differing body habitus and behavioral habits. The United States Preventative Task Force has differing recommendations for weight gain depending on pre-pregnancy BMI, with less weight gain (11 to 20 lb) for obese pregnant mothers and more weight gain (28 to 40 lb) for underweight pregnant mothers [61]. The recommended amount of weight gain during pregnancy was associated with increased *Bifidobacterium* compared to those with excessive weight gain [7]. When controlling for pre-pregnancy BMI, there were no differences in diversity with excess gestational weight gain [8,62]. There have been changes in bacterial abundance with excess gestational weight gain. Increased gestational weight gain was associated with decreased *Prevotella* and *Dialister* in the third trimester [8]. *Dialister* has been associated with insulin sensitivity [7]. For those with normal pre-pregnancy BMI, when they had excessive gestational weight gain, there was an increase in Firmicutes and Bacteroidetes phyla [62], but in those who were obese prior to pregnancy and who had excessive weight gain, there was an increase only in Bacteroidetes [62]. This could be due to the pre-existence of high quantities of Firmicutes in the context of obesity.

Milk and placental microbiomes also appear to be altered by gestational weight gain during pregnancy. Multiple studies have found that with increased gestational weight gain, there was an increase in milk microbiome alpha diversity [63,64]. Independent of maternal obesity, increased gestational weight gain has been associated with an increased abundance of *Staphylococcus* [43,44,45], decreased *Streptococcus* [64], and decreased *Bifidobacterium* [44,64]. *Staphylococcus* has been associated with the pro-inflammatory state of obesity [65] and may contribute to the infant’s risk of future metabolic diseases. Additionally, the placental microbiome in mothers delivers preterm clusters through excess gestational weight gain [41]. Those with excessive gestational weight gain have decreased species richness, decreased Proteobacteria, increased Actinobacteria, increased Firmicutes, and increased Cyanobacteria [41]. These findings are thought-provoking, as excessive gestational weight gain and decreased placental species richness have both been associated with an increased risk of preterm birth [66], and thus provide a potential mechanism that could be amenable to intervention.

Increased gestational weight gain during pregnancy has been associated with changes in the offspring microbiome from the neonatal period to adulthood. Gestational weight gain has been associated with decreased *Akkermansia* abundance at 1 month [27], increased diversity at 6 months [27], increased enrichment of microbial glucose and glycogen degradation pathways, and increased microbial phenylalanine, cysteine/serine, folate, thiamin, biotin, and pyridoxine synthesis pathways at 8 months [67]. These studies suggest that gestational weight gain impacts the infant’s gut bacterial role in energy storage, which could have implications on their risk of obesity and metabolic syndrome. Another study examining women aged 19 to 44 years old whose mothers demonstrated excess gestational weight gain found that those who were exposed to excess gestational weight gain had increased visceral adiposity and increased fecal *Acidaminooccus*, a bacteria associated with adiposity, in adulthood [68]. These findings suggest that adiposity, even in adulthood, may have fetal and microbial origins.

### 2.4. Body Composition

Given that pre-pregnancy BMI is a flawed proxy for adiposity [69], nutritional scientists are moving toward the use of body composition. Body composition examines fat mass (brown, subcutaneous, visceral fat) versus fat-free mass (muscle, organs, or bone), or skeletal muscle mass. Body composition can be studied using various technologies that vary in cost and resolution. Increased white visceral adipose tissue during pregnancy releases pro-inflammatory cytokines and free fatty acids, which can alter the epigenome of the fetus’ muscle, liver, and adipose tissue [14]. These adaptations can increase the risk of developing metabolic syndrome and nonalcoholic fatty liver disease in children. As mentioned previously, mothers with increased BMI have also been found to have significant differences in their HMO profile [22,27], but no studies have directly examined the relationship between maternal body composition and HMO concentrations.

Given the release of free fatty acids and the changes in the HMO composition of the mother’s milk, there are likely alterations in the maternal and infant microbiome. However, this has not been studied well. In a study examining adults with type 2 diabetes, adults with a greater lean tissue index (lean tissue mass divided by height squared) had a higher ratio of Firmicutes to Bacteroidetes phylum [70]. In a rat model with a high-protein diet versus a fat and sucrose diet, a high-protein diet was associated with decreased fat mass, increased alpha diversity, and increased abundance of *Lactobacillacea* and *Bifidobacterium* in the mother [71]. In the offspring of the high-protein mice, they found a decrease in fat mass in both male and female mice, but sex-dependent differences in the microbiome [71]. Both male and female offspring showed differences in beta diversity, but in males, there was also increased alpha diversity, increased *Bifidobacterium*, increased *Muribaculaceae*, and decreased *Lachnospiraceae* [71]. In an observational study examining 140 pregnant mothers, increased fat mass during pregnancy was positively associated with increased *Akkermansia*, *Blautia*, and *Bilophila* [8]. *Bilophila* is a bile-resistant bacillus that expands in the presence of dietary fats and has been associated with increased intestinal inflammation in a mouse model [72]. There have been no other studies to date examining maternal body composition during pregnancy and the microbiome, and further research is needed to define the role of body composition on the microbiome and to better understand the microbial mechanisms of maternal and neonatal metabolic changes.

### 2.5. Gestational Diabetes

Gestational diabetes is one of the most common complications during pregnancy and its prevalence has been increasing alongside both the obesity epidemic and increasing maternal age at the time of conception [73]. Although gestational diabetes has been associated with elevated pre-pregnancy BMI, it has also been found to occur independently of body habitus [73]. Factors such as family history, parity, multiple pregnancies, and genetic factors may also increase the mother’s risk [73]. Gestational diabetes is characterized by a hyperglycemic state during pregnancy. To meet the demands of the developing fetus, a state of transient hyperinsulinism is necessary to store energy; however, some mothers are unable to compensate for hyperinsulinism and develop hyperglycemia due to pancreatic beta cell dysfunction [74]. There is evidence that an altered microbiome with decreased short-chain fatty acid-producing bacteria, decreased amino acid-degrading bacteria, increased Firmicutes to Bacteroidetes ratio, and increased Gram-negative bacteria lead to gut inflammation, gut permeability, increased dyslipidemia, and insulin resistance [75]. Additionally, a Western diet (low fruits and vegetables, high sodium, and fat) was associated with an increased risk of gestational diabetes, whereas a Mediterranean diet (higher bread, cereal, legume, fish, and olive oil diet) was associated with a decreased risk of gestational diabetes [76].

The maternal microbiome is altered in the setting of gestational diabetes. One study found that the vaginal, oral, and intestinal microbiomes were distinctly different from the non-diabetic microbiome in Bray–Curtis distance analyses comparing compositional similarity [15]. They found that the oral cavity had more Proteobacteria and fewer Firmicutes in gestational diabetes, but there were no significant abundance differences in the intestinal or vaginal microbiome [15]. In contrast, one study examining the vaginal microbiomes of 502 pregnant persons found that gestational diabetes was associated with vaginal dysbiosis [16]. Regarding the gut microbiome, some experts argue that the origin of gestational diabetes is microbial [70,72,73,74]. When comparing the gut microbiota of pregnant mothers in the first trimester, those who developed gestational diabetes had increased Ruminococcaceae family [17], butyrate-producing bacteria *Faecalibacterium* [77], and *Eubacterium* [77]. In the third trimester of pregnancy, gestational diabetes is associated with increased gut *Bacteroides* [78], *Streptococcus*, and Enterobacteriaceae family [77]. All of these bacteria are associated with gut inflammation [79]. Further studies are necessary to determine whether altered gut bacteria in the third trimester predict gestational diabetes earlier than the glucose tolerance test in the second trimester.

Gestational diabetes also alters the microbial signatures of the infants of affected mothers. In one study, Chinese infants of mothers with gestational diabetes and normal BMI were found to have decreased alpha diversity of meconium, as well as altered Firmicutes [28] and Proteobacteria at the phylum level [28]. Another study examining meconium from a similar population of mothers with gestational diabetes found that the genus level altered *Prevotella* [15], *Streptococcus* [15], *Bacteroides* [15], and *Lactobacillus* abundances [15]. *Lactobacilus* is important for the de novo synthesis of amino acids [70]. The decreased abundance of *Lactobacillus* may have implications for protein metabolism in newborns.

It is difficult to disentangle the role of pre-pregnancy BMI and gestational diabetes on the microbiome as many studies are underpowered to adjust for BMI or do not include more nuanced assessments of body composition. However, one study comparing the maternal gut microbiome in mothers with gestational diabetes to those who were normoglycemic controlled for pre-pregnancy BMI found decreased *Clostridium* and *Veillonella* after controlling for pre-pregnancy BMI [29]. Another study examining maternal milk microbiome changes with gestational diabetes and pre-pregnancy BMI demonstrated microbial differences after adjusting for gestational diabetes [46]. Nonetheless, when examining the role of gestational diabetes in the milk microbiome of mothers without obesity, they did not find significant differences [46]. Another study examining the infant meconium microbiome of infants born to mothers with gestational diabetes and healthy mothers found gestational diabetes to be a driver of Bacteroidetes, Firmicutes, and Proteobacteria after adjusting for first-trimester maternal BMI [18]. Alterations in Bacteroides have been associated with type 2 diabetes [74]. Furthermore, another study found that neonates of mothers with gestational diabetes had decreased *Lactobacillus*, *Flavonifractor*, *Erysipelotrichaceae*, and *Gammoproteobacteria* after adjusting for pre-pregnancy BMI [30]. These findings provide evidence that gestational diabetes independently alters the microbiomes of mothers and infants.

## 3. Maternal Diet and the Microbiome

There is evidence to suggest that maternal diet may play a critical role in shaping the microbiome during pregnancy and in neonates independent of maternal body habitus. Maternal dietary intake during pregnancy is associated with maternal gut, vaginal, and milk microbiome composition [31,39,80]. Subsequently, the neonatal microbiome is influenced by the maternal diet. Potential mechanisms of transfer to the infant include vaginal delivery, the placenta, or the amniotic fluid. The effects of the maternal diet on the infant stool microbiome persist after delivery for at least 6 weeks [32] and have been found to be greater among infants delivered vaginally than those delivered via cesarean section [31]. However, when examining the drivers of the infant microbiome up until 6 months, chest-feeding status was the primary driver rather than maternal dietary intake [19].

Fat intake is associated with microbial shifts during pregnancy. Saturated fatty acid intake has been positively associated with the gut microbial Simpson diversity index in obese/overweight participants [8]. In normal-weight subjects, increased monounsaturated and polyunsaturated fatty acids are associated with *Ruminococcus* and *Paraprevotella* abundance [8]. *Ruminococcus* has been previously positively associated with polyunsaturated fatty acid supplementation and plant-based diets [7,81]. In a study examining mother-infant pairs in the Mediterranean, maternal lipid intake has been associated with decreased Bacteroidetes and increased Firmicutes relative abundance prior to delivery, consistent with findings in obesity [31]. Regarding the milk microbiome, an increased intake of saturated fatty acids and monounsaturated fatty acids was inversely related to the relative abundance of Corneybacterium in American mothers [47]. However, the study was unable to examine the lipid profiles of the milk itself to see if this affected the milk fat composition [8]. Fat intake has also been directly associated with changes in the infant microbiome [32]. A maternal diet high in total lipids, saturated fatty acids, and mono-unsaturated fatty acids has been associated with the enrichment of the Firmicutes phylum and depletion of the Proteobacteria phylum in infant meconium [31]. Infants of mothers with high-fat intake during pregnancy were found to have lower *Bacteroides* that persisted from birth to 6 weeks of age [32]. This effect was not modified by pre-pregnancy BMI or gestational diabetes, suggesting that maternal diet may be a primary maternal driver of the infant microbiome.

Fruit and vegetable consumption also influences the maternal microbiome. Particularly, a Mediterranean diet with higher plants and limited animal protein appears to be influential [82,83,84]. Mothers who consume vegetarian diets have lower relative abundances of *Collinsella*, *Holdemania*, and *Eubacterium* but an increased abundance of *Roseburia* and *Lachnospiraceae* compared to their omnivorous counterparts in the gut microbiome during the second trimester of pregnancy [82]. *Lachnospiraceae* breaks down polysaccharides into short-chain fatty acids and has been associated with people who practice a vegetarian diet [28,85]. A study examining adherence to a Mediterranean diet throughout pregnancy in Hawaiian mothers found increased maternal gut microbiome diversity and an increased abundance of bacteria that produce short-chain fatty acids [83]. A predominantly plant/fish protein diet also alters the milk microbiome. In a primate study providing a “Mediterranean diet” compared to a “Western diet” (high animal protein, high sodium, and high sugar), they found that the mammary tissues had a 10-fold higher abundance of *Lactobacillus* with the Mediterranean diet [84]. A potential mechanism for these alterations in the milk microbiome may be via the entero-mammary pathway, where gut bacteria are transmitted to the mammary gland by dendritic cells.

Maternal fruit and vegetable intake has been frequently associated with infant gut microbial changes. When examining infant meconium, maternal dietary fiber and vegetable protein intake are negatively associated with the relative abundance of *Coprococcus*, *Blautia*, *Roseburia*, *Ruminococcaceae*, and Lachnospiraceae families [31]. This suggests a more positive microbial profile, as *Blautia* has been associated with increased visceral adiposity in adults [86]. In a study of 39 two-month-old infants in Taiwan, mothers with high fruit and vegetable consumption had a higher abundance of Propionibacteriales, Propionibacteriacea, *Cutibacterium*, Tannerellaceae, *Parabacteroides*, and *Lactococcus* [33]. In contrast, infants of mothers who ate fewer fruits and vegetables had a higher abundance of *Prevotella*, *Isobaculum*, Clostridia, Clostridiales, Lachospiraceae, *Hungatella*, *Lachnoclostridium*, Ruminococcacaea, *flavonifractor*, *erysipelatoclostridium,* Acidaminococcaceae, *Phascolarctobacterium*, *Megamonas*, Betaproteobacteriales, Burkholderiacea, and Suterella [33]. *Cutibacterium* has been found to degrade hexoses to produce propionate [30]. Propionate consumption has been shown to be associated with less antigen presentation on dendritic cells associated with allergic disease in mouse models [87]. Another study found similar results at 6 weeks of age, but found the effect of fruit intake to be modified by mode of delivery [88]; infants born by cesarean section whose mothers had a high fruit intake had increased odds of high *Streptococcus* and *Clostridium*. As the infant ages, there are more environmental drivers to the gut microbiome, but maternal dietary intake appears to continue to play a role. One study found that at 6 months, when controlling for type of milk (mother’s milk versus formula), solid food introduction, mode of delivery, age, maternal education, and race/ethnicity, infants of mothers who ate more fruits and vegetables had increased *Lactobacillus* [19]. *Lactobacillus* has been associated with cellular immunity in infants and has been utilized as a probiotic supplement in atopic diseases with some success [89,90].

Additionally, fish and animal protein sources have been associated with changes in the maternal and infant gut microbiomes. Animal protein intake during pregnancy is positively associated with the maternal gut Shannon diversity index [8]. Low processed meat intake is positively associated with *Lactobacillus* abundance in the gut [19] and vagina [39], and total animal protein intake has been positively associated with *Collinsella* abundance [8]. Regarding the infant microbiome, one study found that higher maternal animal protein intake is associated with a higher abundance of *Veillonella*, *Escherichia/Shigella*, *Klebsiellla*, and *Clostridium* in infant meconium [31]. These bacteria have been associated with infant gut dysbiosis and inflammation in preterm infants [91,92,93,94]. Increased maternal fish and seafood intake is positively associated with increased *Streptococcus* in six-week-old infants regardless of the mode of delivery or maternal BMI [31].

Carbohydrates have also been associated with alterations in maternal and infant microbiomes. Increased carbohydrates during pregnancy have been associated with increased Bacteroidetes in the maternal gut microbiome prior to delivery [31]. Increased total carbohydrates and sugars during pregnancy were associated with improved vaginal health with a lower Nugent score [39]. Regarding the milk microbiome, increased total carbohydrates, disaccharides, and lactose were negatively associated with an abundance of Firmicutes in lactating mothers in the United States [47]. There is minimal information regarding maternal diet and the milk microbiome, and further research is needed to characterize the microbial differences.

## 4. Future Directions—Dietary and Probiotic Interventions

Given the impacts of maternal nutritional status and diet on the maternal and infant gut microbiome, there have been efforts to improve maternal and neonatal health outcomes through prebiotic, probiotic, and dietary interventions. Unfortunately, these interventions have varying levels of success.

Multiple randomized control trials utilizing probiotics have been conducted in overweight pregnant mothers. There have been seven studies to date that have examined the use of probiotics in the prevention of gestational diabetes, and a recent meta-analysis found that probiotics had no effect on the risk of gestational diabetes, cesarean section, gestational weight gain during pregnancy, or large for gestational age infants [95]. Another study found that fish oil and/or probiotic supplementation to overweight pregnant mothers in early pregnancy to assess gestational weight gain and body composition had no significant differences in gestational weight gain or body composition [96], but supplementation decreased *Ureaplasma* and *Prevotella* [97]. A recent meta-analysis found that increased *Ureaplasma* abundance is associated with preterm rupture of membranes, preterm birth, chorioamnionitis, and bronchopulmonary dysplasia, but the evidence is low-quality [98]. Thus, probiotics that decrease the abundance of *Ureaplasma* may have benefits for pregnancy, but further large, randomized control trials are necessary [99]. To date, there is little evidence to support the use of probiotics during pregnancy.

Dietary and exercise interventions during pregnancy have been performed with variable success. In an animal model, methyl-donor nutrients (folate, vitamin B12, choline, methionine, and betadine) provided during pregnancy and lactation to mice receiving a high-fat diet led to decreased cytokine expression and decreased colonic vitamin D receptor (*VDR*) signaling in pups [100]. VDR signaling affects vitamin D metabolism [82]. The UPBEAT trial enrolled obese pregnant mothers to participate in a low glycemic index diet plus physical activity, which resulted in decreased skinfold thickness, gestational weight gain, improved metabolome in mothers, and decreased infant subscapular skinfold thickness z-score at 6 months [101]. Another study examined if the use of a “HealthyMoms” smartphone app during pregnancy would improve gestational weight gain, glycemia, and insulin resistance [102], and did not find any significant differences in clinical outcomes. However, the mothers did have improved healthy eating scores post-partum [83]. Further research is needed to examine the microbial changes in the mother and neonate following nutritional interventions.

## 5. Gaps in the Literature

Currently, the majority of studies examining maternal nutritional factors, such as BMI, body composition, diet, and maternal/infant microbiomes, are descriptive as they utilize 16S rRNA sequencing. Multi-omic studies evaluating metabolomic and microbiome metagenomics would allow for a more thorough understanding of the interactions between host metabolism, the microbiome, and microbial metabolism and function. Longitudinal studies from pre-pregnancy and early pregnancy through the period of exclusive chest-feeding would allow for a thorough characterization of interactions between maternal health, the maternal microbiome, and infant health and microbiome. Another large gap in the literature is the effect of maternal undernutrition, including macronutrient and micronutrient deficiencies, low BMI, and inadequate gestational weight gain in both the maternal and infant microbiomes. Additionally, current methods for assessing maternal diet are inadequate as 24 h recalls, food frequency questionnaires, and other survey methods may not accurately capture dietary intake and quality. As the double burden of malnutrition continues to increase globally among pregnant people, a clear understanding of the effects of malnutrition on the microbiome and infant outcomes is necessary to identify novel targets for intervention to optimize outcomes and development among offspring.

## 6. Conclusions

Maternal metabolic factors regarding adiposity, lean mass accretion, insulin resistance, gestational weight gain, and diet all have microbial implications for the mother, the milk produced, and the offspring. Dietary differences during pregnancy appear to be one of the largest drivers of the maternal microbiome but are the most difficult to study reliably due to the methodology utilized. There also is a paucity of studies on the implications of undernutrition on the maternal microbiome. Microbial shifts in mothers and their offspring can increase their risk of future metabolic diseases. Although interventions with probiotics during pregnancy have not been successful, dietary changes appear to be the most promising. Further research efforts should concentrate on multi-omic approaches and utilize dietary assessments throughout pregnancy and the lactation period, the mother’s milk composition, and the maternal and neonatal microbiome in vulnerable populations to provide precise nutritional and microbiome-directed interventions.

## Figures and Tables

**Figure 1 microorganisms-11-01569-f001:**
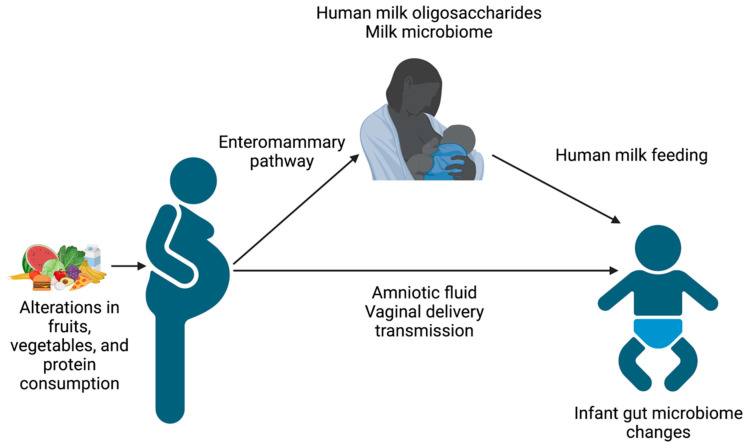
Schematic of the pathways by which maternal nutrition influences the infant gut microbiome. Figure created using Biorender.com, accessed on 31 March 2023.

**Figure 2 microorganisms-11-01569-f002:**
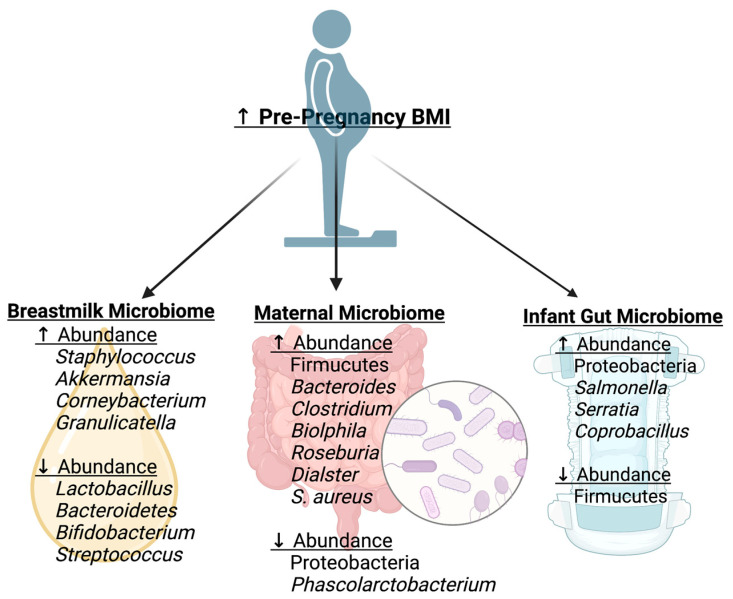
Elevated pre-pregnancy body mass index (BMI) and associated microbial changes. Elevated pre-pregnancy BMI is associated with unique alterations in bacterial abundance in the breast milk microbiome, maternal gut microbiome, and infant gut microbiome. Figure was created using Biorender.com, accessed on 2 June 2023.

**Table 1 microorganisms-11-01569-t001:** Maternal intestinal microbiome changes due to various anthropometric and nutritional states. References are listed in brackets.

Maternal Factor	Maternal Gut Microbiome Diversity	Maternal Gut Microbiome Increased Abundance	Maternal Gut Microbiome Decreased Abundance
Elevated Pre-Pregnancy BMI	Decreased diversity [6]	Firmicutes [6], *Bacteroides* [7], Clostridium [7], *S. aureus* [7] *Biolphila* [8], *Roseburia* [8], and *Dialster* [8]	Proteobacteria [6] and *Phascolarctobacterium* [8]
Underweight Pre-pregnancy BMI	Decreased [9,10,11]	*Acidaminococcus* [12]	Firmicutes [13] and Bacteroidetes [13]
Excessive Gestational Weight Gain		*Prevotella* [8], *Dialister* [8], Firmicutes [14], and *Bacteroidetes* [14]	*Bifidobacterium* [7]
Gestational Diabetes		Ruminococcaceae family [15], *Faecalibacterium* [16], *Eubacterium* [16], *Streptococcus* [16], Enterobacteriaceae family [16], and *Bacteroides* [17]	
Increased Fat Intake	Increased Simpson diversity [8]	*Ruminococcus* [7,8], *Paraprevotella* [8]	*Bacteroidetes* and Firmicutes [18]
Increased Vegetable Intake		*Roseburia* [19] and *Lachnospiraceae* [19]	*Collinsella* [19], *Holdemania,* and *Eubacterium* [19]
Increased Animal Protein Intake	Increased Shannon diversity [8]	*Collinsella* [8]	
Increased Carbohydrate Intake		*Bacteroidetes* [18]	

**Table 2 microorganisms-11-01569-t002:** Infant gut microbiome changes due to various anthropometric and nutritional states. References are listed in brackets.

Maternal Factor	Infant Gut Microbiome Diversity	Infant Gut Microbiome Increased Abundance	Infant Gut Microbiome Decreased Abundance
Maternal Elevated Pre-pregnancy BMI	Increased [27]	Proteobacteria [6] Vaginal delivery infants [9]: *Bacteroides fragilis*, *Escherichia coli*, *Veillonella dispar*, *Staphylococcus*, and *Enterococcus*	Firmicutes [6]
Maternal Gestational Weight Gain	Increased [27]		*Akkermansia* [27]
Maternal Gestational Diabetes	Decreased [28]	*Streptococcus* [15], Firmicutes [28], *Clostridium* [29], *Veillonella* [29], and	*Bacteroides* [15], *Lactobacillus* [15,30], Proteobacteria [28], *Flavonifractor* [30], *Erysipelotrichaceae* [30], and *Gammoproteobacteria* [30]
Maternal Increased Fat Intake		Firmicutes [31]	Proteobacteria [31] and *Bacteroides* [32]
Maternal Increased Fruit and Vegetable Intake		*Lactobacillus* [19], Propionibacteriales [33], Priopionibacteriaceae [33], *Cutibacterium* [33], Tannerellaceae [33], *Parabacteroides* [33], and *Lactococcus* [33]	*Coprococcus* [31], *Blautia* [31], *Roseburia* [31], *Rumiococcaceae* [31], and *Lachnospiracea* [31]
Maternal Increased Animal Protein Intake		*Veillonella* [31], *Escherichia/Shigella* [31], *Klebsiella* [31], and *Clostridium* [31]	

## Data Availability

No new data were created or analyzed in this study. Data sharing is not applicable in this article.

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
