# Peer review of "Maternal Nutritional Status and the Microbiome across the Pregnancy and the Post-Partum Period"

_microorganisms, 2023, doi:10.3390/microorganisms11061569_

Round 1

Reviewer 1 Report

The authors reviewed the alterations in gut microbiomes of parent and their infant by parental body conditions and parental diet during pregnancy and post-partum period. The manuscript is well written and the topic is sound, which might provide important suggestions to pregnant women. However, there are few issues need to be solved.  

1.     What is the mean of four trimesters? Please check and explain it carefully. Actually, body parental condition is key factor for microbiomes, which should be inserted in the title.

2.     Both tables are poorly drawn. The authors should draw them clearly and concisely.

3.     The authors indeed presented the effects of many source of parent diet such as fat, vegetable, fruit, fish and animal protein source, and carbohydrates on parental and infant gut microbiomes, while the authors should compile the information of exact malnutrition effects on microbiomes.

Author Response

Changes made to the text are highlighted in yellow on the file “Microorganisms-2350358-Reviewer1 changes”

  1. What is the mean of four trimesters? Please check and explain it carefully. Actually, body parental condition is key factor for microbiomes, which should be inserted in the title.

Thank you for this question. The fourth trimester being utilized in field of Obstetrics in Gynecology to acknowledge the physiologic transition in the mother after the first 12 weeks after delivery.

Mehta A, Srinivas SK. The Fourth Trimester: 12 Weeks Is Not Enough. Obstet Gynecol. 2021 May 1;137(5):779-781. doi: 10.1097/AOG.0000000000004373. PMID: 33831929.

Body parental condition is a bit of a more general term and may not adequately reflect the specific focus of nutritional status in the pregnancy and post-partum period. To address this and improve clarity for readers who may not be familiar with the term, we have revised our title to “Maternal nutritional status in the pregnancy and post-partum period.”

  1. Both tables are poorly drawn. The authors should draw them clearly and concisely.

Thank you for the feedback on the tables.

Regarding Table 1, we clarified the maternal factors to make it more clear on how they differ per Reviewer 2 comments. In column 2, we removed the words “potential” and added “Maternal gut microbiome”. In columns 3 and 4, we added “Maternal Gut Microbiome” for clarity.

Regarding Table 2, maternal was added to the first column for clarity. The last column regarding functional roles was removed due to a paucity of data.

  1. The authors indeed presented the effects of many sources of parent diet such as fat, vegetable, fruit, fish and animal protein source, and carbohydrates on parental and infant gut microbiomes, while the authors should compile the information of exact malnutrition effects on microbiomes.

It is difficult to say the exact effects of malnutrition on the microbiome. Most studies examining the maternal microbiomes and nutritional status are descriptive using 16S rRNA sequencing, rather than shotgun metagenomic sequencing. There is also very little information regarding metabolomics which would assess activity and function of the identified microbiome constituents. To address this comment, we have added this limitation to the Gaps in the Literature section.

Reviewer 2 Report

This review paper primarily explores the influences of maternal malnutrition on the composition of microbiomes in maternal organs (such as the gut, vagina, placenta, and breast milk), as well as their impact on the fetal gut microbiota. Furthermore, the article comprehensively reviews the effects of maternal diet on both maternal and fetal gut microbiomes, highlighting the crucial role of dietary components in shaping the microbial composition of both the mother and the fetus. Based in these research progress, this review provides insights for future precise interventions targeting the microbiomes of pregnant mothers and fetuses. And it also highlighting the current limitations and future directions of the related research.

Comments:

1.     It is recommended to place the reference numbers for relevant citations before the period at the end of the respective sentences.

2.     Consider replacing the term "parental" with "maternal" to emphasize the focus on mothers.

3.     The name of the items in Table 1 and Table 2 should be made consistent whenever possible to ensure uniformity. For example, it was “Diversity” in table 1, while it was “Infant Gut Microbiome Diversity” in table 2.

4.     The categories of Maternal Factors in the table 1 and table 2 display some content overlap. It is suggested to reorganize and classify the relevant maternal nutritional factors accordingly.

5.     Similar to the comment above, the factors discussed in "PRE-PREGNANCY BMI AND ITS ROLE IN MICROBIAL CHANGES" exhibit overlapping relationships, such as “OVERWEIGHT/OBESITY”, “GESTATIONAL WEIGHT GAIN” and “BODY COMPOSITION”. The categorization is not clear enough and does not fully align with the title of “PRE-PREGNANCY BMI”. It is recommended to reclassify them accordingly in both the text and the corresponding tables.

6.     The paper introduces some contradictory research findings or inconsistent conclusions but does not thoroughly discuss the possible underlying factors. It is suggested to conduct a comprehensive analysis and discussion of the relevant content.

7.     While the paper emphasizes the impact of nutritional status and dietary composition on the microbiomes in the mother and fetus, the definite function of the microbiomes in both the mother and fetus should be adequately addressed, which will give the review more significance.

Although the English language and style are fine, certain sentences could still need to be improved.

Author Response

Changes made to the text are highlighted in yellow on the file “Microorganisms-2350358-Reviewer2 changes”

This review paper primarily explores the influences of maternal malnutrition on the composition of microbiomes in maternal organs (such as the gut, vagina, placenta, and breast milk), as well as their impact on the fetal gut microbiota. Furthermore, the article comprehensively reviews the effects of maternal diet on both maternal and fetal gut microbiomes, highlighting the crucial role of dietary components in shaping the microbial composition of both the mother and the fetus. Based in these research progress, this review provides insights for future precise interventions targeting the microbiomes of pregnant mothers and fetuses. And it also highlighting the current limitations and future directions of the related research.

Comments:

  1. It is recommended to place the reference numbers for relevant citations before the period at the end of the respective sentences.

We have revised our manuscript formatting to reflect this request.

  1. Consider replacing the term "parental" with "maternal" to emphasize the focus on mothers.

This was altered per reviewer request.

  1. The name of the items in Table 1 and Table 2 should be made consistent whenever possible to ensure uniformity. For example, it was “Diversity” in table 1, while it was “Infant Gut Microbiome Diversity” in table 2.

Thank you for noticing these inconsistencies. The tables were altered accordingly.

  1. The categories of Maternal Factors in the table 1 and table 2 display some content overlap. It is suggested to reorganize and classify the relevant maternal nutritional factors accordingly.

Thank you for this comment. While we recognize that while these maternal factors may be associated with pre-pregnancy BMI, they are independent factors of metabolic health. BMI cannot distinguish lean body mass from fat mass. Gestational weight gain can differ based off of obstetrician recommendations and behavioral changes. Gestational diabetes is also dependent on maternal age and genetic pre-disposition, in addition to metabolic health.

  1. Similar to the comment above, the factors discussed in "PRE-PREGNANCY BMI AND ITS ROLE IN MICROBIAL CHANGES" exhibit overlapping relationships, such as “OVERWEIGHT/OBESITY”, “GESTATIONAL WEIGHT GAIN” and “BODY COMPOSITION”. The categorization is not clear enough and does not fully align with the title of “PRE-PREGNANCY BMI”. It is recommended to reclassify them accordingly in both the text and the corresponding tables.

We agree that the section title “Pre-pregnancy BMI and its role in microbial changes” may not be best. To address the reviewer’s concerns, we changed the section title to “Maternal metabolic health and its role in microbial changes”. However, we would argue that gestational weight gain, body composition, and gestational diabetes are not necessarily associated with pre-pregnancy BMI. These conditions often have some overlap though each is a distinct condition/state with differing definitions.  BMI cannot distinguish lean body mass from fat mass. Gestational weight gain can differ based off of obstetrician recommendations and behavioral changes. Gestational diabetes is also dependent on maternal age and genetic pre-disposition, in addition to metabolic health.  We have attempted to make this more clear in “Gestational weight gain” paragraph 1, “Body Composition” paragraph 1, and “Gestational diabetes” paragraph 1.

  1. The paper introduces some contradictory research findings or inconsistent conclusions but does not thoroughly discuss the possible underlying factors. It is suggested to conduct a comprehensive analysis and discussion of the relevant content.

Thank you for your thoughtful remark. We have tried to expand on our discussion of contradictory research. See “Overweight/Obesity” paragraphs 1-4 and “Gestational weight gain” paragraph 1. These additions are highlighted in the text. 

  1. While the paper emphasizes the impact of nutritional status and dietary composition on the microbiomes in the mother and fetus, the definite function of the microbiomes in both the mother and fetus should be adequately addressed, which will give the review more significance.

We agree that understanding the functional role of the bacteria would strengthen the review. Thus, we have tried to include this information when available (see Obesity paragraph 4, Gestational weight gain paragraph 3, Gestational Diabetes paragraphs 1&2, Maternal Diet paragraph 3). Because the studies are largely descriptive and do not include metagenomic sequencing and host metabolomics, we are unable to comment on exact mechanisms between maternal nutritional status and the microbiome. There is also a paucity of information regarding metabolomics. To address this comment, we have added this limitation to the Gaps in the Literature section.

Reviewer 3 Report

Proper nutrition during pregnancy and post-partum is crucial for both parents and their offspring. Inadequate or excessive nutrition can impact the gut microbiomes of parents and infants, affecting the risk of obesity and metabolic diseases. This review examines changes in the gut, vaginal, placental, and milk microbiomes of parents in relation to pre-pregnancy BMI, gestational factors, and diet. It also explores the impact on the infant's gut microbiome. Dietary differences significantly influence the parent's microbiome and subsequently the milk and offspring microbiomes. This review is timely and interesting as more and more papers focused on the role of gut microbiota in pregnant and their offspring. I only have some minor comments for this review.

1.     In terms of the section ‘PRE-PREGNANCY BMI AND ITS ROLE IN MICROBIAL CHANGES’, it would be beneficial to see all those factors in a graph.

2.     Most of the microbiome studies are associated research, which only showed the associations between health and microbial. I think the casualty studies are needed in future studies. Please add more details in ‘GAPS IN THE LITERATURE’ section.

Author Response

Proper nutrition during pregnancy and post-partum is crucial for both parents and their offspring. Inadequate or excessive nutrition can impact the gut microbiomes of parents and infants, affecting the risk of obesity and metabolic diseases. This review examines changes in the gut, vaginal, placental, and milk microbiomes of parents in relation to pre-pregnancy BMI, gestational factors, and diet. It also explores the impact on the infant's gut microbiome. Dietary differences significantly influence the parent's microbiome and subsequently the milk and offspring microbiomes. This review is timely and interesting as more and more papers focused on the role of gut microbiota in pregnant and their offspring. I only have some minor comments for this review.

  1. In terms of the section ‘PRE-PREGNANCY BMI AND ITS ROLE IN MICROBIAL CHANGES’, it would be beneficial to see all those factors in a graph.

Thank you for your suggestion. We have revised the title of this section of the manuscript to more accurately reflect the content presented. We have also attempted to revise and include these maternal factors in Table 1. Additionally, we have developed a figure (Figure 2) to more clearly present the regarding increased pre-pregnancy BMI and associations with microbial changes.

  1. Most of the microbiome studies are associated research, which only showed the associations between health and microbial. I think the casualty studies are needed in future studies. Please add more details in ‘GAPS IN THE LITERATURE’ section.

We strongly agree with the reviewer that this represents a large gap the literature. We have now added this to the gaps in the literature section outlining potential mechanisms to address this with a multi-omic approach.

Round 2

Reviewer 2 Report

    The manuscript effectively explores the influence of maternal nutritional imbalances on the composition of microbial communities in various maternal organs, including vagina, placenta, and breast milk, as well as their significant impact on the fetal gut microbiota. The authors have provided a thorough analysis of the effects of maternal diet on both maternal and fetal gut microbial communities, emphasizing the pivotal role of dietary components in shaping the microbial composition of both the mother and the fetus. Furthermore, it highlights the current limitations in the field and proposes future research directions, thus providing a valuable resource for researchers and practitioners interested in this area. In conclusion, the manuscript offers valuable insights into the influence of maternal nutritional imbalances on microbial communities during pregnancy. The well-structured and informative content, coupled with the identification of research gaps and future directions, makes it a valuable contribution to the field. Thanks for the authors' efforts in addressing my comments during the last revision process.